# Cotranslational microRNA mediated messenger RNA destabilization

Trinh To Tat[1], Patricia A Maroney[1], Sangpen Chamnongpol[2], Jeff Coller[1], Timothy W Nilsen[1]*

[1]Center for RNA Molecular Biology, Case Western Reserve University, Cleveland, United States; [2]Affymetrix, Inc., Cleveland, United States

**Abstract** MicroRNAs are small (22 nucleotide) regulatory molecules that play important roles in a wide variety of biological processes. These RNAs, which bind to targeted mRNAs via limited base pairing interactions, act to reduce protein production from those mRNAs. Considerable evidence indicates that miRNAs destabilize targeted mRNAs by recruiting enzymes that function in normal mRNA decay and mRNA degradation is widely thought to occur when mRNAs are in a ribosome free state. Nevertheless, when examined, miRNA targeted mRNAs are invariably found to be polysome associated; observations that appear to be at face value incompatible with a simple decay model. Here, we provide evidence that turnover of miRNA-targeted mRNAs occurs while they are being translated. Cotranslational mRNA degradation is initiated by decapping and proceeds 5' to 3' behind the last translating ribosome. These results provide an explanation for a long standing mystery in the miRNA field.

## Introduction

microRNAs, a large family of regulatory molecules discovered over twenty years ago, have been shown to exert pervasive effects on a wide array of biological processes (reviewed in *Ameres and Zamore, 2013*; *Ebert and Sharp, 2012*). It has become clear that a majority of mRNAs are targeted by miRNAs via limited base pairing interactions between the miRNAs and miRNA responsive elements (MREs) (*Bartel, 2009*). Although there are exceptions, most MREs are located in 3' UTRs (*Bartel, 2009*). The current consensus view of miRNA-mediated gene regulation is that it involves post-transcriptional down regulation of protein production from targeted mRNAs. Repression by miRNAs is largely attributable to mRNA destabilization and several lines of evidence indicate that this destabilization involves recruitment of factors which participate in normal mRNA decay pathways (*Bagga et al., 2005*; *Guo et al., 2010*; *Eichhorn et al., 2014*; *Behm-Ansmant et al., 2006a*; *2006b*; *Eulalio et al., 2008b*; *Jonas and Izaurralde, 2015*). These factors include deadenylases and decapping enzymes (e.g. *Braun et al., 2011*; *Chekulaeva et al., 2011*; *Rehwinkel et al., 2005*).

Although much has been learned regarding the biological functions of mRNAs and their mechanism(s) of action, there is one aspect that has largely eluded explanation; i.e. the interplay between the translation machinery and miRNA-mediated repression. In very early studies, the first miRNA targeted mRNAs were found to be fully associated with polysomes (*Olsen and Ambros, 1999*; *Seggerson et al., 2002*; *Wightman et al., 1993*). These observations have been reproduced numerous times over the years; whenever the subcellular localization of mRNA targets has been examined, those mRNAs are found to be exclusively on polysomes and unambiguous evidence has indicated that these polysomes are engaged in active translation (*Gu et al., 2009*; *Nottrott et al., 2006*; *Petersen et al., 2006*). Complementing these studies are several studies indicating that a substantive fraction of miRNAs themselves is also associated with actively translating ribosomes (*Maroney et al., 2006*; *Nelson et al., 2004*; *Kim et al., 2004*). Other observations, including the

*For correspondence: twn@case.edu

**eLife digest** DNA encodes instructions to make proteins. The DNA is first copied to make molecules of messenger ribonucleic acid (mRNA) that are then "translated" into proteins by large particles known as ribosomes. MicroRNAs are a type of very small RNA molecule that can reduce the amount of protein produced from mRNAs in animals and other eukaryotic organisms. However, the mechanism by which microRNAs achieve this has been unclear.

Many groups of researchers have shown that microRNAs promote the degradation of particular mRNAs. Others have shown that the mRNAs that are targeted by microRNAs are generally bound to active ribosomes. Since the degradation of mRNAs is widely believed to occur away from the ribosomes, these two sets of observations have been considered to be incompatible with each other. Tat et al. set out to resolve this paradox by studying how microRNAs work in fruit fly cells.

The experiments showed that microRNAs do indeed promote the degradation of the mRNAs they bind to and that these mRNAs are exclusively associated with active ribosomes. Furthermore, this process uses the same cellular machinery that is used for the normal destruction of mRNAs. MicroRNAs help to recruit this machinery to their target mRNAs and thereby enhance mRNA break down.

Tat et al.'s findings provide an explanation for a longstanding puzzle in microRNA research. However, although this mechanism is widely used, it does not appear to apply to all mRNAs targeted by microRNAs, so a future challenge is to understand how these other mRNAs are broken down.

fact that MREs are less effective when present in open reading frames and the presence of a so-called ribosome shadow (absence of MREs in the 3' UTR just downstream of stop codons) are consistent with an interaction between miRNA-effector complexes and translating ribosomes (*Bartel, 2009*). In sum, this body of evidence has led to the development of the hypothesis, which still persists in the literature, that miRNAs regulate translation in some way during the elongation phase but this hypothesis has been recalcitrant to mechanistic proof.

It is difficult to reconcile the large body of evidence that shows that miRNA targeted mRNAs are associated with translating ribosomes with the equally compelling body of evidence that indicates that most miRNA-mediated downregulation occurs through mRNA destabilization. The conundrum is that the prevalent view of mRNA decay holds that decay occurs when mRNAs are in a ribosome free state, perhaps in subcellular aggregates known as P bodies, sites of concentrated RNA degradative activities (reviewed in *Valencia-Sanchez et al., 2006*). Here, we provide a solution to this longstanding puzzle by demonstrating that miRNA-mediated mRNA decay occurs cotranslationlly. Decay is initiated by mRNA decapping while targeted mRNAs are polysome bound and proceeds 5' to 3' following the last translating ribosome. These observations provide a unifying explanation for a large amount of what has been thought to be disparate experimental data.

## Results

We are not aware of any study in which both miRNA targeted mRNA stability and subcellular localization (i.e. polysome association) were analyzed. To address this gap and to avoid potential pitfalls associated with transient transfection approaches, we elected to study stable cell lines expressing miRNA targeted mRNAs and relied on endogenous miRNAs to repress them. Because miRNA targeted mRNAs are particularly well characterized in *Drosophila* (*Stark et al., 2003*), we chose to analyze *Drosophila* mRNAs encoding the proapoptotic proteins, reaper and hid. The reaper mRNA contains one microRNA recognition element (MRE) for the *Drosophila* miR-2 family of miRNA (miRs 2a, 2b, 2c, 6, 11, 13a, 13b, 308) in its 3'UTR (*Figure 1A*, *Figure 1—figure supplement 1*) (*Jovanovic and Hengartner, 2006*). The hid mRNA contains five MREs for the miRNA, bantam, in its 3' UTR (*Figure 1A*, *Figure 1—figure supplement 1*) (*Brennecke et al., 2003*). To avoid the issue that overexpression of either the reaper or hid proteins would promote apopotosis, the entire 3' UTRs of both mRNAs were appended to the firefly luciferase coding sequence and 5' UTR (*Figure 1—figure supplement 1*). Two versions of each 3' UTR, one containing wild type MREs and one

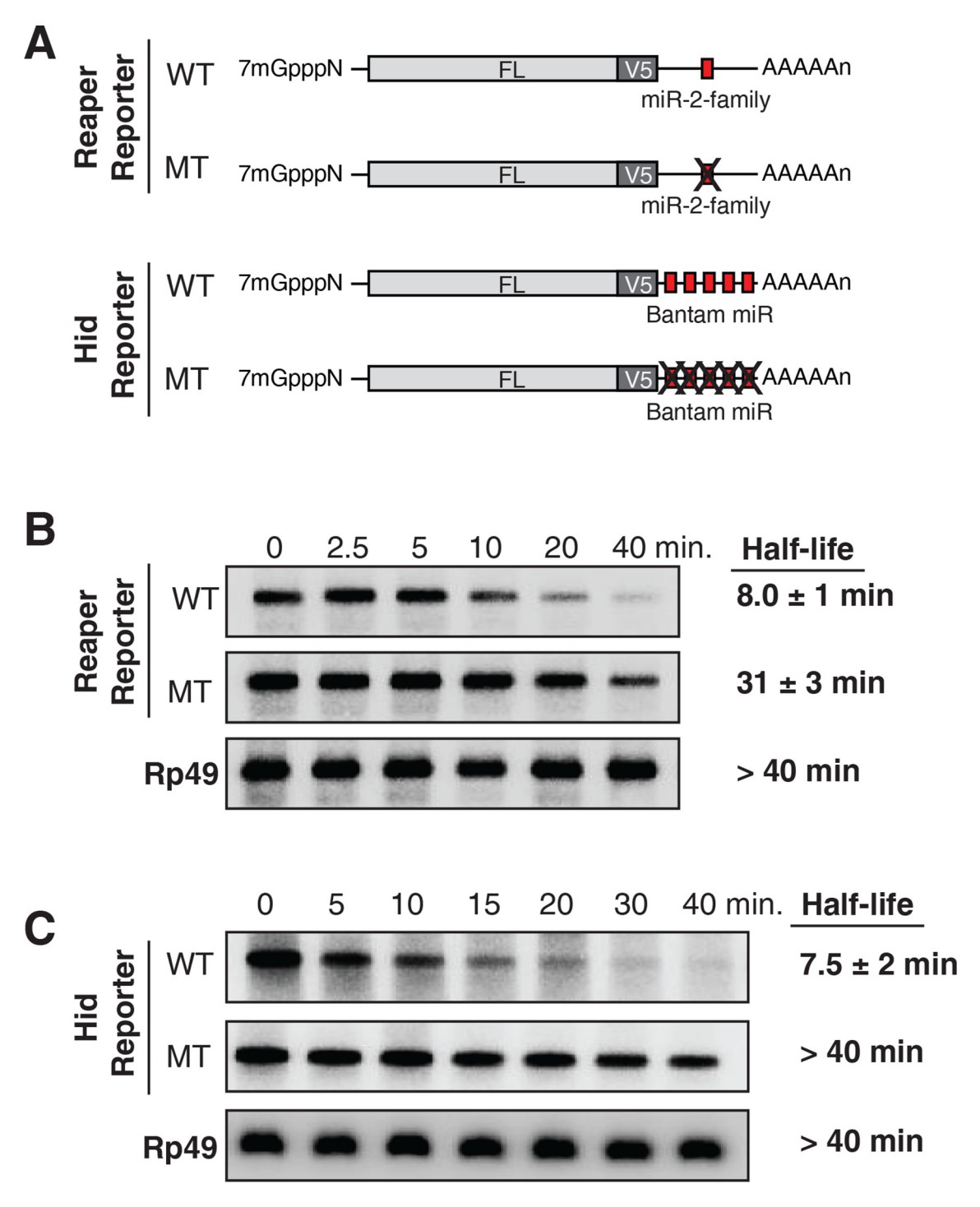

**Figure 1.** Destabilization of reaper and hid reporter mRNAs by endogenous miRNAs. (**A**) Schematic of reporter mRNAs analyzed; WT denotes mRNAs with wild type MREs and MT denotes mRNAs where the MREs were inactivated by mutation. FL is the firefly luciferase coding region and V5 is the V5 epitope tag. (**B**) Stability of reaper reporter mRNAs containing wild type (WT) or mutant (MT) MREs. Cells were treated with actinomycin D for the indicated times; total RNA was prepared and analyzed by Northern blotting using a probe to the firefly luciferase open reading frame. Rp49 denotes the mRNA encoding ribosomal protein L32, a highly stable mRNA, which served as an internal loading control. (**C**) Stability of hid reporter mRNAs containing wild type (WT) or mutant (MT) MREs. All procedures were exactly as in (**B**).

The following figure supplements are available for figure 1:

*Figure 1 continued*

**Figure supplement 1.** Sequences of 3' UTRs of reaper and hid with wild type and mutant MREs highlighted.
**Figure supplement 2.** miRNA mimics complementary to mutant MREs destabilize reaper and hid reporter mRNAs.

containing mutant MREs were used in the construction of four stable *Drosophila* S2 cell lines. Cell lines were judged to be stable when no loss of cells was observed in the presence of the drugs used for their selection and when luciferase levels remained constant over several cell doublings. Reaper constructs were under control of the inducible metallothionine promoter (*Bunch et al., 1988*) while the hid constructs were under control of the constitutive actin promoter (*Angelichio et al., 1991*).

Because experimental evidence indicates that mRNA destabilization is at least one component of miRNA-mediated repression in *Drosophila* (reviewed in *Huntzinger and Izaurralde, 2011*; *Izaurralde, 2015*; *Eulalio et al., 2008a*), we measured the stability of each of the four mRNAs. We first determined the half life of reaper reporter mRNAs. After transcriptional shut off, the reaper reporter mRNA with the wild type MRE decayed with a half life of approximately 8 min (*Figure 1B*). The half life of the reaper reporter mRNA containing the mutant MRE was approximately 31 min, about four fold longer than the mRNA with the wild type MRE (*Figure 1B*).

To confirm that this effect on mRNA stability was miRNA-mediated we designed a double stranded miRNA mimic. When loaded into an Ago protein the targeting strand contained bases complementary to the nucleotides changed in the mutant reaper MRE. While a control mimic had no effect on the half life of the reaper reporter containing the mutant MRE the mimic complementary to the mutant MRE markedly reduced the half life of the reporter to a level comparable to that of the reaper reporter containing the wild type MRE (*Figure 1—figure supplement 2*).

To determine if the four fold effect on mRNA stability this was the maximal miRNA-mediated effect possible, expression of the reaper reporter mRNA was induced either ten-fold or one hundred fold with different concentrations of copper and half lives were again determined. Importantly the reaper reporter mRNA containing the wild type MRE had the same half life at all levels of expression; a result which indicated that miR-2 family miRNAs were well in excess of the targeted mRNA and that the reaper MRE was likely to be fully occupied at steady state (data not shown). Therefore a four fold enhancement of decay was the maximum effect that could be exerted by microRNAs on this mRNA target.

Parallel analysis of the hid constructs revealed a similar microRNA-mediated effect on mRNA stability (*Figure 1C*). The hid reporter mRNA with five wild type MREs had a half life of approximately 7.5 min whereas the mRNA with the mutant MREs had a half life longer than forty minutes (*Figure 1C*). A Bantam mimic analogous to the miR2 mimic described above caused marked destabilization of the reporter containing mutant Bantam sites, confirming that the observed effect on mRNA stability was miRNA-mediated (*Figure 1—figure supplement 2*). We concluded from those experiments that for these two targeted mRNAs, endogenous miRNAs caused substantive destabilization.

We then assessed the subcellular localization of all four mRNAs using sedimentation velocity sucrose gradients and Northern blotting (*Figure 2*). Remarkably, fractionation of cytoplasmic extracts in this manner revealed that all detectable mRNAs for all four of the constructs cosedimented with polysomes, an observation that suggested that the mRNAs were associated with translating ribosomes (*Figure 2B* and *Figure 2—figure supplement 1*). To ensure that polysome associated mRNA accounted for most or all mRNAs present in the cell, we performed an "accounting" experiment analogous to that we previously reported for miRNAs (*Maroney et al., 2006*) We found that the vast majority of reporter mRNAs was indeed associated with polysomes and that there was negligible mRNA loss during extract preparation or gradient fractionation (*Figure 2—figure supplement 2*) compare lanes total and gradient. To determine if polysome associated mRNAs were actually being translated, cells were treated with either emetine, an irreversible inhibitor of translation elongation (*Akinboye and Bakare, 2011*), or Harringtonine, an inhibitor of protein synthesis that binds to the A site of initiating ribosomes only and thereby prevents formation of the first peptide bond (*Fresno et al., 1977*). Because Harringtonine inhibits formation of only the first peptide bond; ribosomes that are already elongating continue to translocate through the open reading

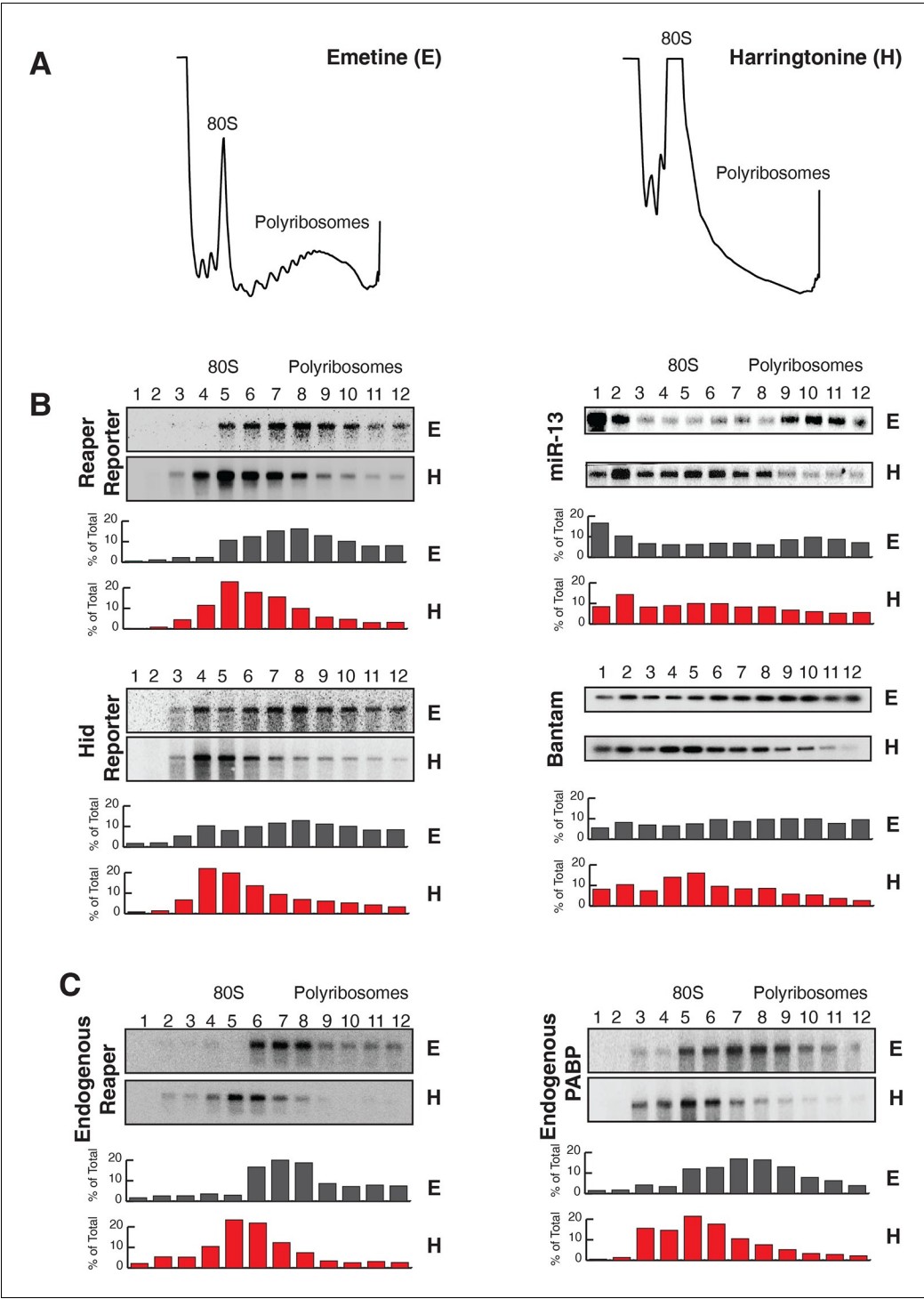

**Figure 2.** Reaper and hid reporter mRNAs containing wild type MREs are present on actively translating ribosomes. (A) Representative UV absorbance (254 nm) traces of polysome gradients on which either cytoplasmic extracts from emetine (E) treated or harringtonine (H) treated cells were fractionated. (B) Northern blots and quantitation of signal from fractions of polysome gradients as in (A); Blots from cells expressing reaper and hid reporter mRNAs containing wild type MREs are on the left. Sedimentation profiles, as assessed by splinted ligation, and signal quantitation of endogenous mir-13b and bantam miRNAs are shown on the right. (C) Northern blot and quantitation of signal from fractions of polysome gradients in which extract of cells treated with emetine (E) or harringtonine (H) were sedimented. The probes were to either endogenous reaper or to polyA binding protein (PABP) mRNAs.

The following figure supplements are available for figure 2:

*Figure 2 continued on next page*

*Figure 2 continued*

**Figure supplement 1.** Reporter mRNAs containing mutant MREs are present on translating ribosomes.
**Figure supplement 2.** First indication of cotranslational decay of reaper reporter mRNA containing a wild type MRE.
**Figure supplement 3.** Most mRNAs are accounted for in polysome gradients.
**Figure supplement 4.** Reporter mRNAs containing wild type MREs are associated with lighter polysomes than reporter mRNAs containing mutant MREs.

frame and eventually terminate and disassociate from the mRNA. As elongating ribosomes clear the mRNA, polysomes become smaller and the mRNA shifts to lighter fractions of the gradient; newly initiating messages accumulate in the monosome region (80S). Extracts prepared from Harringtonine treated cells were sedimented on sucrose gradients and fractions were analyzed by Northern blotting. These analyses revealed that all four mRNAs shifted to lighter fractions; a result which indicated that all four mRNAs were associated with actively translating ribosomes (*Figure 2B* and *Figure 2—figure supplement 1*). Consistent with other earlier observations (*Maroney et al., 2006*; *Nelson et al., 2004*) (*Kim et al., 2004*), both endogenous bantam and a representative member of the miR-2 family (miR-13b) showed similar sedimentation behavior as these targeted mRNAs; i.e. a substantive amount of each miRNA sedimented with polysomes and shifted to lighter fractions upon Harringtonine treatment (*Figure 2B*). Endogenous mRNAs encoding the ribosomal protein L32 (RP49) and polyA binding protein (PABP) were, as expected, also present on translating ribosomes (*Figure 2C*). At this point in our analysis it was clear that endogenous miRNAs were causing marked destabilization of targeted mRNAs, yet all detectable mRNAs were associated with actively translating ribosomes.

How was microRNA-mediated enhanced mRNA decay compatible with active translation of the targeted mRNAs? We considered two possible scenarios. In the first, translational initiation was somehow blocked on targeted mRNAs. After elongating ribosomes cleared the mRNA, the ribosome free mRNA was degraded so quickly that it could not be detected. The second scenario was that decay was occurring while the mRNAs were being translated; i.e. in a ribosome bound state. A clue to distinguish between these possibilities came from examination of long exposures of the Northern blot across polysome gradients shown in *Figure 2*. We observed a family of hybridizing species that were smaller than full length (*Figure 2—figure supplement 3*). We speculated that these species might have resulted from partial degradation of the mRNA. If this was the case, we predicted that such fragments would become enriched relative to full length mRNA if new synthesis of mRNA was prevented. Accordingly, we stopped ongoing transcription with Actinomycin D and incubated cells for approximately one mRNA half life in order to allow for partial decay of pre-existing mRNA; translation was then arrested with emetine (*Figure 3B*).

Cells expressing the reaper reporter mRNA containing the wild type MRE were treated this way and extracts were fractionated on sucrose gradients (*Figure 3—figure supplement 1*). Upon analysis by Northern blotting, we observed a striking accumulation of truncated mRNA fragments (*Figure 3C*). These fragments were detected with a probe which hybridized to the 3' region of the open reading frame but were not observed when a probe which hybridized to the 5' region of the reading frame was used (*Figure 3D*). Comparable fragments were greatly underrepresented when the same analysis was performed on the reaper reporter mRNA containing the mutated MRE (*Figure 3C*). We concluded from this that the fragments represented mRNA decay intermediates degraded from the 5' end of the mRNA. Because the decay intermediates were ribosome associated, as indicated by their sedimentation behavior, these data show that enhanced microRNA-mediated mRNA decay occurs cotranslationally.

Parallel analysis was carried out on the hid constructs (*Figure 3—figure supplement 2*). Because the hid 3' UTR was so long (3kb) relative to the coding region the results were less clear cut than with the reaper constructs. Nevertheless, ribosome associated decay fragments derived from mRNAs containing wild type MREs accumulated in great excess to those derived from the mRNA containing mutant MREs.

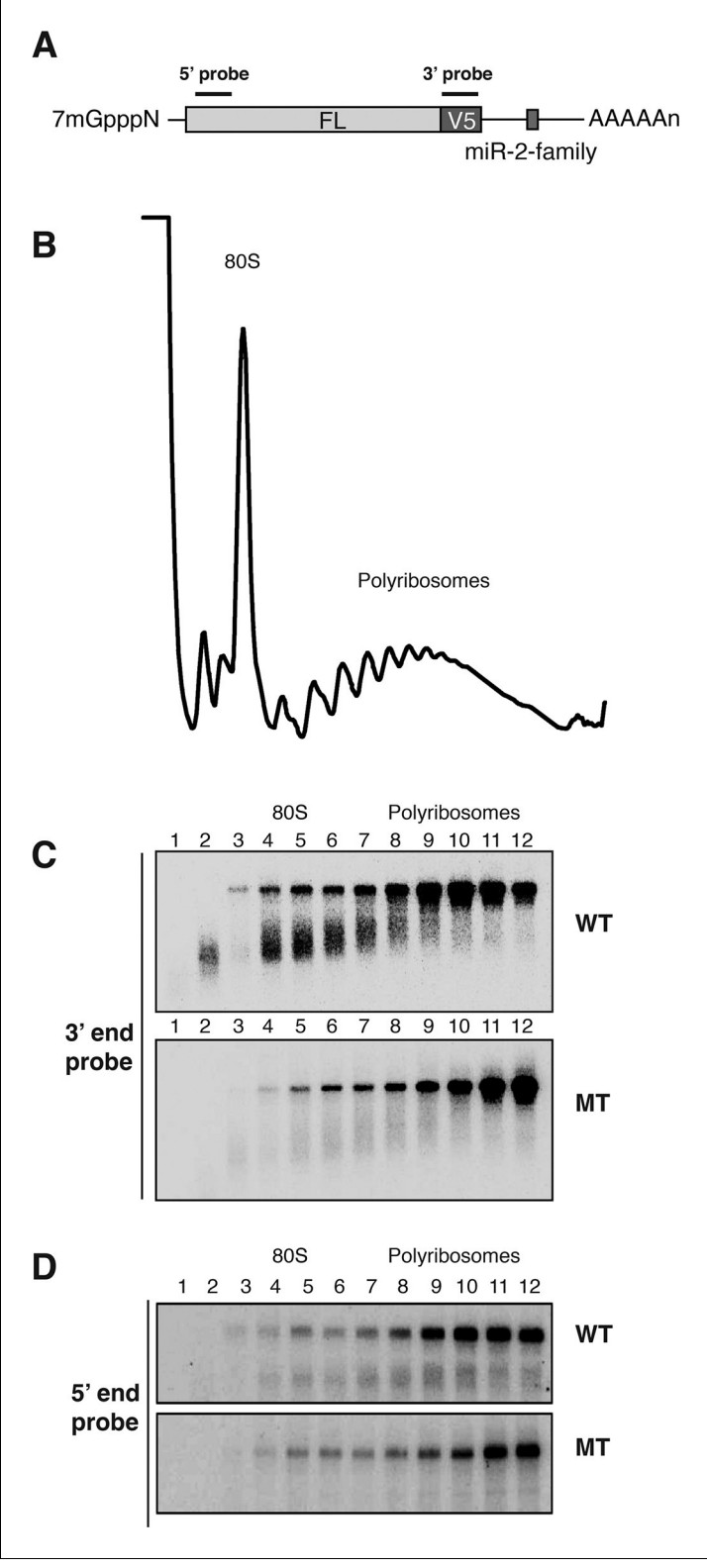

**Figure 3.** Cotranslational decay of reaper reporter mRNA. (**A**) Schematic representation of probes used for detection of reaper reporter mRNA. (**B**) Representative UV absorbance trace (254 nm) of a polysome gradient fractionating cytoplasmic extract prepared from cells treated with actinomycin D for ten minutes. Actinomycin treatment for this time had no effect on overall protein synthesis as assessed by polysome profiling (*Figure 3—figure supplement 1*). (**C**) Northern blot analyses of fractions from polysome gradients in which cytoplasmic

*Figure 3 continued on next page*

*Figure 3 continued*

extracts from cells expressing reaper reporter mRNAs containing either a wild type (WT) or mutant (MT) MRE were sedimented. Northern blots were probed with a fragment that hybridized to the 3' region of the firefly reporter open reading frame, see (A). (C) The same as in (D) except that the blots were probed with a fragment that hybridized to the 5' region of the firefly luciferase open reading frame, see (A).

The following figure supplements are available for figure 3:

**Figure supplement 1.** 10 min actinomycin D treatment does not affect overall protein synthesis.

**Figure supplement 2.** Evidence for cotranslational mRNA degradation of hid reporter mRNA containing wild type MREs.

---

Having shown that miRNA-mediated mRNA destabilization was cotranslational, we wished to determine the pathway by which this mRNA decay occurred. Because decay intermediates were truncated from the 5' end it seemed likely that they were derived from decapped mRNAs. To determine if decapping was required for decay, the decapping factors DCP1 and 2 were knocked down by RNAi. The reaper reporter mRNA containing the wild type MRE was markedly stabilized in cells wherein decapping was compromised, a result that indicated that removal of the cap was an important step in initiating targeted mRNA degradation (*Figure 5A*).

Several lines of evidence indicate that decapping is contingent upon prior deadenylation of the mRNA (*Braun et al., 2012*; *Eulalio et al., 2009*; *Giraldez et al., 2006*). It has been shown in *Drosophila* that miRNAs bound to Ago1 recruit GW182; a protein that in turn recruits the CCR4-Not deadenylase complex (e.g. *Braun et al., 2011*; *Chekulaeva et al., 2011*). This complex deadenylates mRNA and in some way, perhaps by direct recruitment, facilitates mRNA decapping by Dcp1 and 2 (*Behm-Ansmant et al., 2006a*; *2006b*; *Rehwinkel et al., 2005*). To determine if the cotranslational decay we observed followed this pathway, we examined the adenylation status of miRNA-targeted mRNAs. Using a polyA tail length assay (*Rio et al., 2011*) and *Figure 4A*, we observed what appeared to be "bands" for both reaper reporter and hid reporter tails. Since these "bands" migrated at the size expected for deadenylated mRNAs this result suggested that deadenylated species accumulated in polysome associated mRNAs (*Figure 4B*). To confirm this interpretation the "band" from the reaper reporter was excised and sequenced. The results confirmed this interpretation (*Figure 4C*). In contrast, control non-microRNA targeted mRNA displayed normal polyA tails. The presence of deadenylated mRNAs in polysome fractions strongly suggested that accelerated decapping was a consequence of accelerated deadenylation.

Following decapping, mRNA degradation is catalyzed by the 5'-3' exonuclease XRN1; XRN1 leaves 5' phosphorylated termini. To determine if the cotranslational decay we observed depended upon an enzyme that leaves 5' phosphorylated termini, we used a ligation strategy designed to detect RNAs which terminated in a 5' phosphate (*Figure 5B*). Unlike most splinted ligation assays which require precise juxtaposition of 5' and 3' ends and use DNA ligase as the sealing enzyme, we used a bridge oligonucleotide that brought a "reporter" oligonucleotide with a free 3' hydroxyl into proximity of mRNA fragments with 5'phosphate termini and used RNA ligase 1 as the sealing enzyme (*Figure 5B*).

To validate this assay we sought to identify substrates that would provide both positive and negative controls. We reasoned that a fully capped mRNA could provide the negative control. We chose to analyze the mRNA encoding ribosomal protein L32, RP49 (*Figure 5C*). Because the RP49 mRNA has a very long half life (see *Figure 1*), we anticipated that it would be essentially all capped at steady state. Consistent with this, only trace amounts of ligation products diagnostic of decapped RP49 mRNA were present in total RNA. However, after treatment with tobacco acid pyrophosphatase (TAP), an enzyme that removes the cap, a robust signal for decapped RP49 was obtained (*Figure 5C*). We then used the ligation assay to analyze the reaper reporter mRNA. In this case we set up the assay to detect any 5' phosphorylated mRNA fragments containing a 5' terminal phosphate present in the region of the reaper reporter mRNA ranging from its transcription start site to just downstream of the AUG initiation codon. Here, we observed strong signals in the absence of TAP a result indicating that decapped mRNAs were present at steady state. Only the band

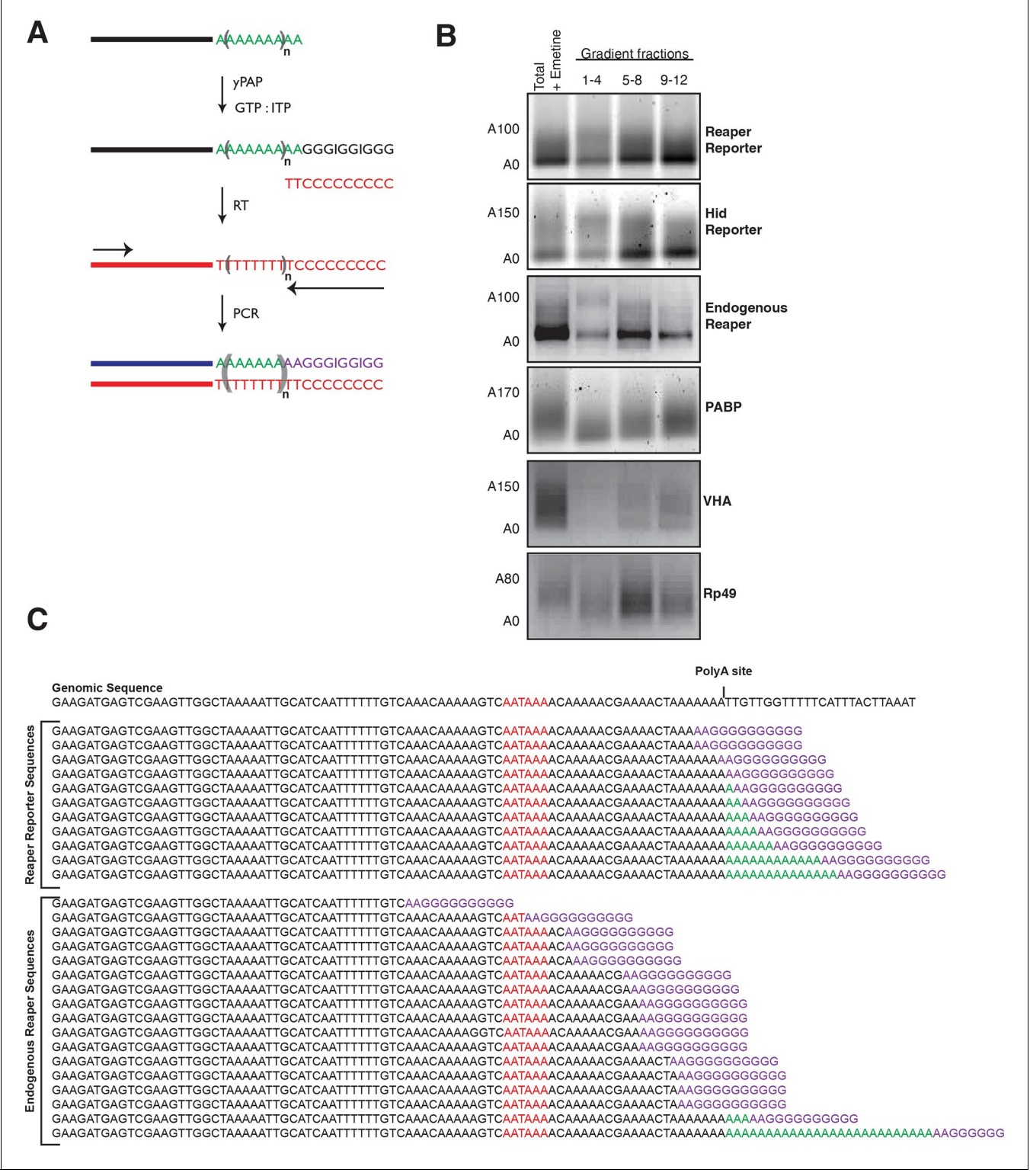

**Figure 4.** Cotranslational miRNA-mediated decay involves mRNA deadenylation. (**A**) Schematic depiction of the polyA tail length assay to measure the length of a polyA tail. (**B**) PolyA tail lengths for the indicated mRNAs. Tail lengths were determined for each mRNA in total RNA and pooled polysome gradient fractions as indicated. This assay does not have the sensitivity to detect the endogenous hid 3′ end. (**C**) The deadenylated species of reaper reporter and endogenous reaper were cloned and sequenced. AATAAA: polyA signal. Genomic DNA sequences are shown in black while the reverse primer is shown in purple.

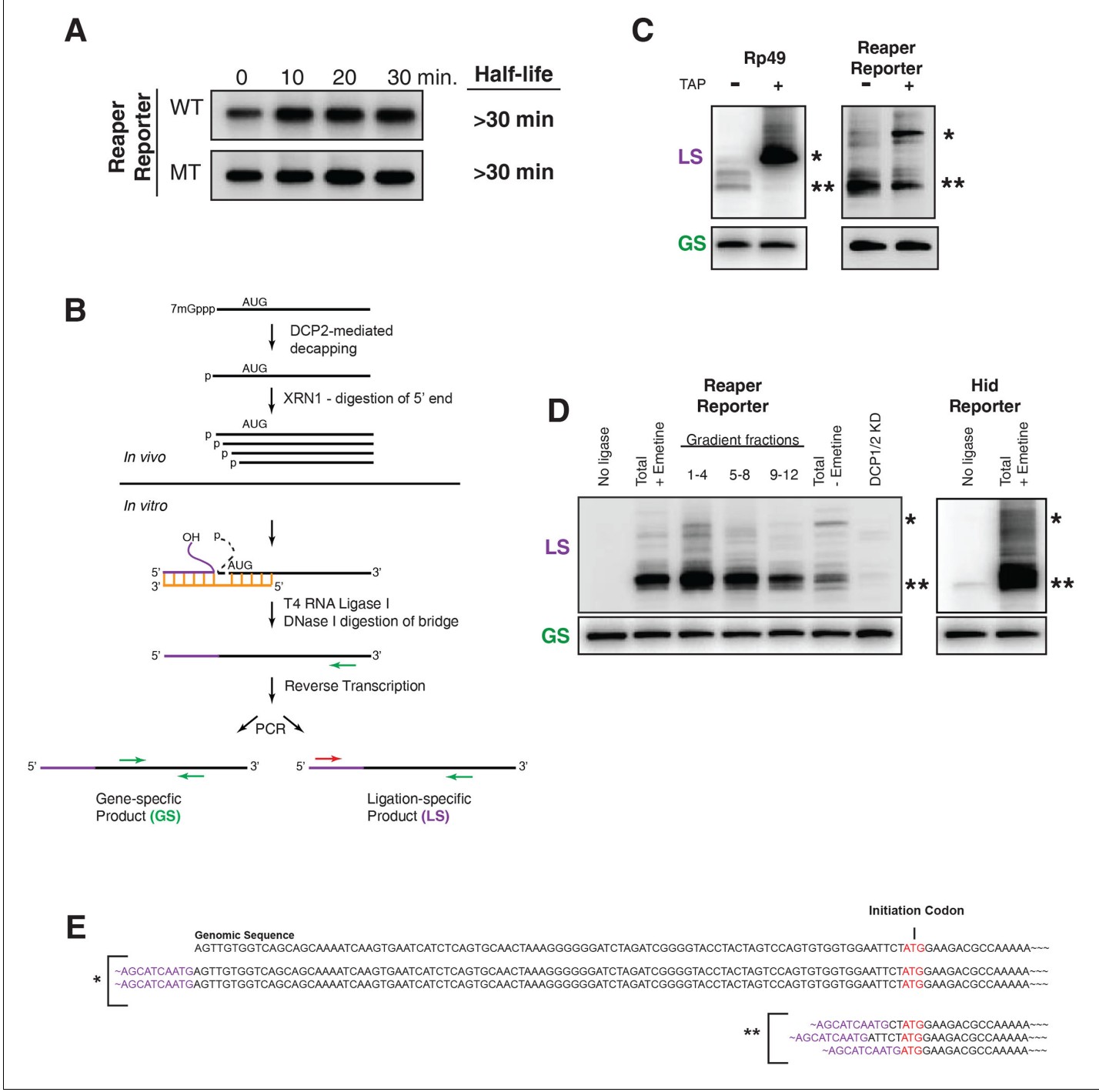

**Figure 5.** Cotranslational miRNA-mediated decay involves mRNA decapping. (A) Stabilization of reaper reporter mRNAs upon knock down of Drosophila Dcp1 and Dcp2. Northern blot analysis of total RNA prepared at the indicated time after addition of actinomycin D and probed with a fragment that hybridized to the 3' region of the firefly luciferase open reading frame. (B) Schematic depiction of the ligation strategy to identify decapped 5' phosphorylated mRNA fragments. (C) Ligation assay for the indicated mRNAs either untreated or treated with tobacco acid pyrophosphatase (TAP). Ligation specific (LS) (decapped RNA) or gene specific (GS) PCR fragments; see (B), Heterogeneity results from digestion by XRN1 following decapping. The single asterisk denotes the position of the product when ligation is at the 5' terminus of the mRNA. The double asterisk denotes ligation products near the AUG initiation codon. (D) Ligations assays on RNA samples from cytoplasmic extract and gradient fractions, or total RNA with or without knock down of decapping enzymes. Labelling is as in (C). (E) Sequencing analysis of the products when ligation is to the 5' terminus of the mRNA and near the ATG initiation codon. Ligation-specific forward primer is shown in purple.

corresponding to the mRNA start site was enhanced by TAP treatment (*Figure 5C*). Importantly, detection of these fragments was ligase dependent and just a trace of fragments were detected in RNA prepared from cells in which DCP 1 and 2 were knocked down (*Figure 5D*). When the assay was used to detect decay fragments across a polysome gradient we observed signal throughout the polysome region of the gradient (*Figure 5C* and see Discussion). Cloning of bands denoted by asterisks showed that their 5' ends mapped at or near the transcription start site and near the translation initiation codon (*Figure 5E*). These results demonstrated that the decay fragments we observed had resulted from decapping of the reporter mRNA while it was polysome bound.

We conclude that cotranslational miRNA mediated mRNA destabilization employs the canonical mRNA decay machinery. This destabilization is accelerated relative to non-targeted mRNAs almost surely due to Ago1 recruitment of GW182 which in turn recruits the deadenylase machinery. Following deadenylation, mRNAs are decapped thereby yielding substrates for XRN1 (*Figure 6*).

## Discussion

We have provided evidence that miRNAs promote enhanced mRNA decay while the mRNA targets are engaged with actively translating ribosomes. While we have analyzed only two miRNA targets, we think it likely that these observations will be broadly relevant. In cases where the subcellular localization of miRNA targets have been studied, they have invariably been found to be polysome associated (*Olsen and Ambros, 1999*; *Gu et al., 2009*; *Seggerson et al., 2002*; *Wightman al., 1993*; *Nottrott et al., 2006*; *Petersen et al., 2006*). Moreover, miRNAs themselves have been shown to be polysome associated (*Maroney et al., 2006*; *Nelson et al., 2004*; *Kim et al., 2004*). Our observations are also consistent with a large body of transcriptome wide analyses obtained from studies primarily in mammalial systems that have shown that mRNA destabilization is the primary if not sole effect of miRNA-mediated down regulation of protein production from targeted mRNAs (*Guo et al., 2010*; *Eichhorn et al, 2014*). They are also consistent with studies in *Drosophila* S2 cells that have shown that miRNA effector complexes as well as general decay enzymes including deadenylases and decapping factors are ribosome associated (*Antic et al., 2015*). While cotranslational miRNA-mediated mRNA decay rationalizes a very large body of published results, there are some observations that are difficult to reconcile with this unifying model of miRNA-mediated action. For example one study in a *C. elegans* cell free system showed that miRNA-targeted mRNAs were deadenylated and translationally repressed but not degraded (*Wu et al., 2010*). Another example, also from *C. elegans,* used ribosome profiling in vivo to show that miRNA mediated regulation in some cases correlated with mRNA abundance but in other cases did not (*Stadler et al., 2012*). Finally, there are also examples of *Drosophila* miRNA targeted mRNAs which appear to be regulated at the level of translation and not stability (e.g. *Huntzinger and Izaurralde, 2011* and *Behm-Ansmant et al., 2006a*; *2006b*). These examples might indicate that some specific mRNPs are subject to distinct regulatory pathways. Future studies may elucidate mRNP specific factors that mediate these effects.

Cotranslational mRNA decay was shown several years ago for a selected group of mRNAs in budding yeast (*Hu et al., 2009*). More recently, studies have shown that such decay is the dominant if not sole pathway for normal mRNA decay in yeast (*Pelechano et al., 2015*). Our findings extend these observations to higher eukaryotes and suggest that cotranslational decay may be a widespread phenomenon.

Our results do not speak specifically to how mRNA decay is triggered. In that regard there has been a longstanding debate in the field as to whether inhibition of translation initiation is a prerequisite for miRNA-mediated enhanced decay (*Bazzini et al., 2012*; *Djuranovic et al., 2012*; *Eichhorn et al., 2014*; *Subtelny et al., 2014*). Indeed over the years a large variety of different inhibition scenarios have been proposed including mRNA sequestration from the translation apparatus and inactivation of an array of initiation factors (e.g. *Valencia-Sanchez et al., 2006*; *Mathonnet et al., 2007*; *Chendrimada et al., 2007*; *Kiriakidou et al., 2007*; *Wang et al., 2008*; *Fukaya et al., 2014*), the latest being eIF4AI and II (*Fukao et al., 2014*; *Meijer et al., 2013*). Unfortunately none of these mechanisms including eIF4AI and II (see *Galicia-Vázquez et al., 2015*) has received sustained experimental support and it is currently unclear how or if initiation of translation is truly impaired in a microRNA-mediated fashion.

Independent of miRNA-mediated mRNA decapping, mRNAs decap in general (*Coller and Parker, 2004*). For general mRNA decapping, translational initiation is also believed to be in

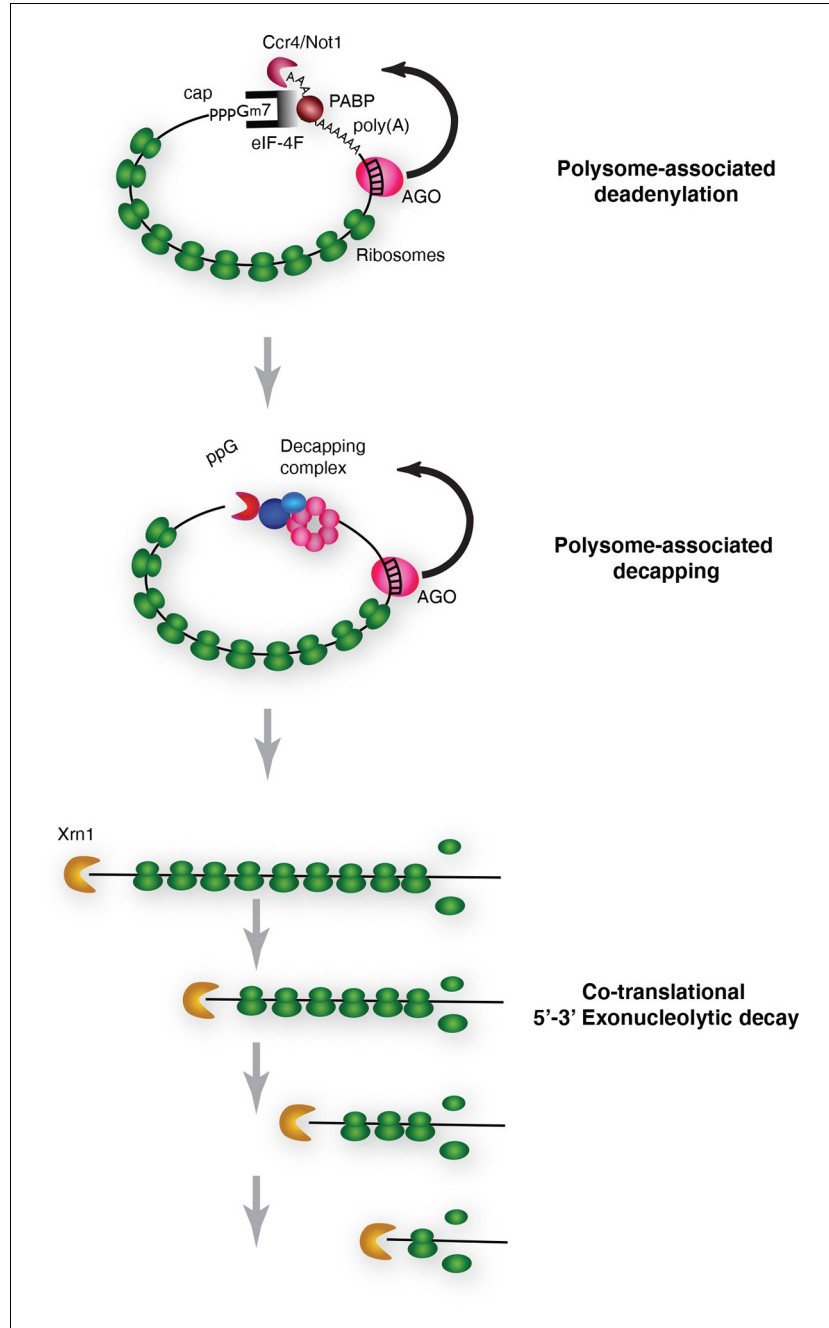

**Figure 6.** Pathway of miRNA-mediated cotranslational mRNA decay. Schematic summary of the model which emerged from the data; for details see the text.

competition with the DCP2/1 holoenzyme. Importantly, despite this well accepted notion that initiation and decapping compete, it remains to be established how, when, or even if dissolution of the translational initiation complex occurs prior to DCP2/1-mediated cleavage. Thus the events triggering decapping are still murky but may include recruitment of the RNA helicase DDX6 (*Chen et al., 2014*; *Mathys et al., 2014*; *Rouya et al., 2014*). Importantly, however, since eIF-4F requires a cap to stimulate translation, decapping itself clearly and dramatically limits any further ribosome association and thus can itself be thought of as a potent and irreversible inhibitor of translation initiation.

The first report of inhibition of initiation of translation came from polysome analysis where it was shown that an miRNA targeted mRNA migrated on lighter polysomes than a control mRNA

(*Pillai et al., 2005*). Notably, however, all of the targeted mRNA was associated with ribosomes; there was no mRNA sedimenting in the RNP region of the gradient (*Pillai et al., 2005*). These results were interpreted to mean that miRNAs repressed translation initiation. We too observe that targeted mRNAs migrate on lighter polysomes than untargeted mRNAs (*Figure 2—figure supplement 4*). However, we suggest an alternative explanation for this phenomenon. It seems possible that the act of decapping itself could account for the altered sedimentation of targeted mRNAs. If decapping is a prerequisite for but not rate limiting for mRNA decay, decapped but not degraded mRNAs could persist. Because already engaged ribosomes would continue to elongate and eventually terminate, decapped but intact mRNAs would shift to lighter polysomes as elongating ribosomes clear the mRNA. We note that this notion is not entirely speculative. We have observed decapped mRNAs with 5' termini that map both near the start site of the mRNA and near the translational initiation codon. Importantly, these RNAs, which would appear to be full length, are detected throughout polysomes of different sizes both large and small (*Figure 5D*). These observations indicate that 5' to 3' exonucleolytic decay is not obligatorily coupled to decapping because if such coupling existed we would not observe full length decapped mRNA. It seems possible the concentration of degradation fragments near the initiation AUG could result from the buildup of ribosomes near the beginning of the open reading frame as observed in ribosome profiling studies (e.g. *Ingolia et al., 2011*). Collectively, these observations suggest that decapping while a prerequisite for mRNA decay is not likely to be rate limiting for decay itself; exonucleolytic digestion by XRN1 is likely to be rate limiting.

In sum, we have provided an explanation for a longstanding mystery in the miRNA field and have suggested a plausible model for miRNA action that is consistent with a large body of experimental results obtained in numerous laboratories.

## Materials and methods

### Oligonucleotides and construct design

All oligonucleotides used in this work were purchased from Integrated DNA Technologies (IDT) and their sequences are shown in *Supplementary file 1*.

PCR products containing full-length 3' UTRs of reaper and hid transcripts were amplified from S2 genomic DNA using overlapping PCR. Oligonucleotide site directed mutagenesis (*Kunkel et al., 1987*) was used to generate the mutant 3' UTR versions containing mutated MREs; one MRE within the reaper 3' UTR and 5 MREs within the hid 3' UTR. Wild type and mutant 3' UTR sequences were cloned downstream of the firefly luciferase (FL) coding sequence into the pMT-V5-HisB vector (Invitrogen, Carlsbad, CA) or pAc5.1-V5-HisB vector (Invitrogen).

### Cell culture and construction of stable cell lines

*Drosophila* S2 cells (ATCC CRL-1963 were obtained from Invitrogen cat no. 10831) were grown in serum-free media Express Five SFM (Gibco) supplemented with 18 mM L-Glutamine and Pen/Strep at 24°C without $CO_2$. Cell density was maintained between $2x10^6$ cells/mL and $8x10^6$ cells/mL.

Stable (S2) cell lines were generated by co-transfection of the experimental constructs (wild-type or mutated MREs) together with a plasmid carrying a hygromycin resistance gene pCoHygro (Invitrogen). 6 mL cells ($2\times10^6$ /mL) were seeded per T25 flask the day before transfection. Cells were transfected using Effectene Transfection Reagent (Qiagen) with 0.5 µg reporter plasmids and 0.5 µg antibiotic resistance plasmids, in DNA-condensation buffer (buffer EC), Enhancer Solution, and Effectene. Medium was changed the next day and the cells were incubated for another 24 hr. After that, hygromycin B (200 µg/ml) (Life Technologies) was added and the cells were incubated in that media for an additional 5 days. In parallel, mock transfected cells were similarly treated and used as controls to monitor cell death. When control cells were no longer alive, the transfected cells were transferred over time into medium containing lower concentrations of hygromycin B (down to 100 µg/ml), until luciferase levels remained constant.

### Drug treatment

Stable cell lines were seeded without hygromycin B at $2x10^6$ cells/mL in T75 flasks. The next day, cells were treated with either 0.1 mg/mL emetine (Sigma-Aldrich) for 10 min, or 2 µg/mL harringtonine (Abcam) for 5 min. Actinomycin D (ActD) (Sigma-Aldrich), a transcriptional inhibitor, was used

at 5 µg/mL to measure mRNA half-lives at indicated time points. For half-life determination, $4x10^6$ cells/mL were treated with 5 µg/mL ActD, and 3.5 mL cells added to 60 mm dishes for each time point. At the indicated times, duplicate 1.5 mL samples were removed, quickly pelleted in a micro-fuge at 1500 xg and resuspended in Tri Reagent (MRC). RNA was extracted, quantitated, and equal amounts were electrophoresed in agarose gels for Northern Blots. In addition to reporters, blots were probed for Rp49 as a loading control. In order to maximize truncated decay fragments, cells were treated with ActD at 5 µg/mL for 1 min (hid) and 10 min (reaper), followed by emetine for 10 min. After drug treatments, cytoplasmic extracts were prepared for further experiments.

## Cell extracts, polysome analysis, and RNA extraction

Cytoplasmic extracts were prepared from a T75 flask containing 20 mL of cells at a density of $8x10^6$ cells/mL. Ten minutes before harvesting, cells were treated with 0.1 mg/mL emetine to irreversibly block protein synthesis during extract preparation. For extracts, cells were swelled in three volumes of hypotonic buffer without detergent (buffer A: 10 mM KCl, 1.5 mM $MgCl_2$, 20 mM Tris pH 7.5), and lysed with 0.2% Triton X100. Extract was spun at 30000 g for 5 min and aliquots ($10A_{260}$ units) of the supernatant were centrifuged through 15%–40% (w/w) sucrose gradients in buffer A at 40000 r.p.m., for 90 min in a Beckman SW41 rotor. After centrifugation, gradients were fractionated from the top by displacement through a flow cell, and $A_{254}$ was monitored with a recording spectrophotometer.

Detailed protocols for RNA extraction are as described (*Rio et al., 2011*). Total RNA was isolated from cell pellets directly homogenized in Tri Reagent (MRC). A 1:5 volume chloroform was added to each sample, followed by vigorous mixing and centrifugation. The aqueous phase was precipitated using isopropanol. After resuspension in buffer containing 1 mM EDTA, 20 mM Tris pH 7.5, and 0.5% SDS (SDS extraction buffer), RNA was reextracted using phenol:chloroform:isoamyl alcohol (25:24:1 v/v/v) and ethanol precipitated. RNA pellets were resuspended in nuclease-free water and concentrations determined using a NanoDrop 2000 (Thermo Scientific).

To prepare RNA from sucrose gradients, fractions were diluted two fold with SDS extraction buffer and digested with 100 µg/mL proteinase K for 10 min at 37°C, followed by extraction with phenol:chloroform:isoamyl alcohol (25:24:1 v/v/v) and precipitation with ethanol and GlycoBlue (Ambion) as carrier.

## Northern blotting

Samples were denatured at 65°C for 8 min in buffer containing MOPS, formamide, and formalde-hyde, and electrophoresed on 1% (reaper) or 0.8% (hid) agarose with MOPS buffer as described (*Rio et al., 2011*).

RNA was transferred from the gel to a nylon membrane (GeneScreen Plus, PerkinElmer) using capillary transfer in transfer buffer (10X SSC) overnight. Transfer membranes were rinsed briefly in 2X SSC and UV crosslinked using a Stratalinker (254 mm; 1200 mJ), followed by baking at 80°C for 1 hr. Membranes were pre-hybridized with a mix containing 15% formamide, 10 mg/mL BSA, 0.5 M sodium phosphate buffer pH 7.0, 1 mM EDTA, 7% SDS with rocking at 42°C (*Rio et al., 2011*). After 2–4 hr, buffer was replaced and probes were added. After overnight hybridization, the membranes were washed in 0.5 X SSC and 0.1% SDS, dried, and visualized by phosphorimaging (Typhoon 9400, GE Healthcare). Images were quantitated using ImageQuant TL 8.1 (GE Healthcare). The details of this protocol are as described (*Rio et al., 2011*).

## Linear PCR to synthesize the $^{32}$P-labeled single-stranded DNA probes

200 nt PCR fragments specific for genes of interest were amplified, gel purified, and quantitated for use as templates. The labeled single-stranded DNA was generated in linear PCR reactions contain-ing $\alpha^{32}$P-dCTP (PerkinElmer) and a strand-specific oligonucleotide primer designed to synthesize a single stranded probe complementary to the RNA target (*Rio et al., 2011*). After PCR, unreacted $\alpha^{32}$P-dCTP was removed with Micro Bio-Spin 6 Chromatography Columns (Biorad).

## RNA interference by double stranded RNAs (dsRNAs)

Expression of the decapping enzymes DCP1 and DCP2 was knocked down using RNA interference (RNAi) following a previously described protocol (*Celotto et al., 2005*). The coding region of the

target of interest was amplified and cloned into pBluescript, flanked by the T3 and T7 promoter sequences. In vitro high-yield synthesis of RNA using T3 and T7 RNA polymerases were carried out to generate single-stranded RNAs (ssRNAs) in both the sense and anti-sense directions. DNaseI (Roche) was added to digest the DNA templates and the ssRNAs were ethanol precipitated. The sense and anti-sense ssRNAs were heated at 95°C for 1 min and annealed in 50 mM NaCl, 20 mM Tris pH 7.5 at 65°C for 10 min followed by slow cooling to room temperature. The dsRNAs were evaluated by electrophoresis on 1% agarose gels and staining with ethidium bromide.

1 mL S2 cells were seeded at $1 \times 10^6$ cells/mL in each well of a 6-well plate. 20 µg of both dsRNAs were added to each well followed by a 30-minute incubation. Fresh medium was added and cells were incubated further at 24°C. Cells were boosted after 3 days of incubation by the addition of fresh dsRNAs (40 µg), followed by an additional 24 hr of incubation.

## Poly(A) tail length assay

The Poly(A) tail length assay uses the method described by Affymetrix/USB with minor revisions (*Rio et al., 2011*). ~40 guanosine (G) nucleotides were added to the 3' end of mRNAs by yeast poly (A) polymerase (Affymetrix/USB). Reactions (12.5 µL) contained 0.5 µg of RNA, 20 mM Tris pH 7.0, 0.6 mM $MnCl_2$, 0.02 mM EDTA, 1 mM ITP, 1 mM GTP, 2 mM $MgCl_2$, 6 U poly(A) polymerase (Affymetrix) and incubated at 37°C for 40 min. Samples were diluted with 50 µL of 10 mM Tris, 1 mM EDTA, 0.5% SDS, pH 7.5. Nucleotides were removed using Micro Bio-Spin 6 Chromatography Columns (Biorad), and RNAs were precipitated with sodium acetate (0.3 M, pH 5.2) in ethanol.

Tailed RNAs were resuspended in water and reverse transcribed using a reverse primer C10T2 designed to anneal within the poly(GI) tail extension. The RNA and primer were heated at 65°C for 5 min, chilled, and then incubated at 42°C with 50 mM Tris-HCl pH 8.3, 75 mM KCl, 3 mM $MgCl_2$, 10 mM dithiothreitol (DTT), 0.5 mM dNTPs, and SuperscriptII (200 U) (Life Technologies). cDNA was amplified using FastTaq polymerase (Roche) in PCR reaction buffer containing additional 0.8 mM $MgCl_2$, 0.2 mM of each dNTP, and primers (0.2 µM each of AnchorC10T2 and a gene-specific forward primer located 100–500 nt 5' to the poly(A) signal). The PCR program was set at 94°C for 6 min, 34 cycles of (94°C for 30 s, 60°C for 30 s, 72°C for 45 s), and then 72°C for 5 min.

## Splinted ligation to detect miRNAs and mRNA phosphorylated 5' termini ends

MicroRNA detection was as described (*Rio et al., 2011*; *Maroney et al., 2007*). The common 5' ligation oligo was 5' end labeled with $\gamma^{32}$P-ATP (PerkinElmer), and 3 pmole were mixed with 1 pmole of bridge oligo and 1 µg total RNA, denatured at 95°C for 1 min and annealed for 2 min at 65°C and 10 min at 37°C. After addition of ligation buffer (final concentration 60 mM Tris pH 7.5, 10 mM $MgCl_2$, 1 mM DTT, 1 mM ATP, 5% (w/v) PEG 6000) and T4 DNA ligase, samples were incubated at room temperature for 1 hr and then digested with 1 u shrimp alkaline phosphatase (Affymetrix) for 20 min at 37°C. Samples were run on 6% denaturing polyacrylamide gels, dried, and visualized with a phosphoimager (Typhoon 9400, GE Healthcare).

5' mRNA termini ligations used an RNA ligation oligo and are illustrated schematically in *Figure 5*. Complementarity to the bridge ended 10 nucleotides from the 3' end of the RNA ligation oligo and started 3 nucleotides 3' to the AUG in the mRNA. Samples in *Figure 5C* were either incubated with or without tobacco acid pyrophosphatase (Promega discontinued product) for an hour at 37°C to remove the $m^7G$ cap and leave 5' phosphate termini.

Annealing conditions were 95°C for 1 min, 65°C for 2 min, and 45°C for 10 min. Ligation buffer and 10 units RNA ligase 1 (NEB) were added and reactions were incubated at 37°C for 1 hr. After that, samples were diluted with 50 mM Tris pH 7.5, 10 mM $MgCl_2$ and digested with DNaseI (Roche) for 15 min at 37°C and precipitated with ethanol.

To detect ligated products, cDNAs with gene specific primers were made starting 192 nt 3' to the AUG. cDNAs were amplified with PCR primers shown schematically in *Figure 5* in reactions containing 5 µCi $\alpha$-$^{32}$P dCTP (PerkinElmer).

PCR products were electrophoresed on 6% native polyacrylamide gels, dried and visualized with a phosphoimager (Typhoon 9400, GE Healthcare).

## Additional information

### Competing interests
TWN: Reviewing editor, *eLife.* The other authors declare that no competing interests exist.

### Funding

| Funder | Grant reference number | Author |
|---|---|---|
| School of Medicine, Case Western Reserve University | INS693040 | Timothy W Nilsen |
| Vietnam Education Foundation | | Trinh To Tat |

The funders had no role in study design, data collection and interpretation, or the decision to submit the work for publication.

### Author contributions
TTT, PAM, Conception and design, Acquisition of data, Analysis and interpretation of data, Drafting or revising the article; SC, JC, TWN, Conception and design, Analysis and interpretation of data, Drafting or revising the article

### Author ORCIDs
Timothy W Nilsen, http://orcid.org/0000-0002-7276-3571

## Additional files

### Supplementary files
• Supplementary file 1. Oligonucleotides used in this study.

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
