## [Decision Letter]

Thank you for submitting your work entitled "Cotranslational microRNA mediated messenger RNA destabilization" for consideration by *eLife*. Your article has been reviewed by three peer reviewers, including Erik Sontheimer and Douglas Black, who is a member of our board of Reviewing editors. The evaluation has been overseen by James Manley as the Senior Editor.

The reviewers have discussed the reviews with one another and the Reviewing Editor has drafted this decision to help you prepare a revised submission.

Summary:

Tat et al. examine the mechanisms of genetic repression by miRNA's, presenting evidence supporting the model that miRNA binding to mRNAs represses expression during active translation on polysomes by inducing mRNA deadenylation and decapping. Reporter genes are constructed containing the reaper and hid 3' UTR's attached to firefly luciferase. These are stably expressed in *Drosophila* S2 cells that contain the endogenous miRNA's miR-2 and Bantam, which target reaper and hid respectively. The authors show that mutating the miR-2 and Bantam miRNA responsive elements (MREs) in these reporters leads to a substantial stabilization of the reporter transcripts. They find that virtually all of the reporter mRNA and a large fraction of the miRNAs are associated with polysomes. The wild type but not the MRE-mutant mRNAs give rise degradation products that are also associated with polysomes and which are missing sequences from their 5' ends. The authors show that the reaper and hid reporters, as well as the endogenous reaper transcript, but not other endogenous transcripts, produce polysomal deadenylated products. Finally, they show using a 5' ligation assay that the reaper and hid reporter transcripts on polysomes are partially decapped and exo-nucleolytically trimmed from their 5' ends. These data support a model where miRNAs lead to mRNA decay during translation, without a prior step of removing the targets from the pool of actively translating mRNAs.

The paper is clearly written and the presented experiments are well executed. A great deal of disparate data has been published on the mechanisms by which miRNAs repress expression of their target mRNAs, with some conclusions seeming to be incompatible. It was agreed by all the reviewers that this work could help clarify important questions in the field. However, the reviewers pointed to several major experiments that were needed to solidify the findings. Questions were also raised regarding how broadly the results could be interpreted.

Essential revisions:

1) The experiments rely on endogenous miRNAs that are thought, but not shown to be present in the S2 cells. The only data linking particular miRNAs to the effects observed are mutations in the miRNA binding sites. These mutations could alter a variety of 3' UTR mediated regulatory processes. The authors need to demonstrate that miR-2 and Bantam mediate all the decay processes observed by blocking their action with antagomirs.

2) The arguments in the paper depend on all the mRNA being on polysomes and thus being the major contributor to the mRNA half-life measured on total RNA. The authors should confirm this by measuring how much mRNA is lost prior to the polysome gradient and how much is pelleted in these gradients. Is there a pool of mRNA sequestered in high molecular weight structures (such as P-bodies) that is not seen on the polysomes but nevertheless contributes to decay rate measurements?

3) The differential decay of the wild type and mutant RNAs is observed in different stable cell lines for each reporter. This could arise from variation in these cell lines affecting mRNA stability. The authors should assess the stability of reporter and endogenous transcripts in multiple clones of each reporter to control for this. Alternatively, both mutant and wild-type reporters could be measured in the same cell line.

4) The paper is written from the point of view that there will be one pathway of translational repression and decay induced by miRNAs. The authors should discuss the possibility that different mRNAs could show different pathways, or the different mechanisms could be balanced differently in different systems. Such a discussion would include several references already given, but also papers by Wu et al. 2010 (PMID:21095586) and Stadler et al. 2012 (PMID:22855835), showing that in some cases mRNA abundance can be stable while protein expression is reduced by a miRNA. In support of cotranslational deadenylation and decapping, they might also reference Antic et al. 2015 (PMID:25918245).

---

## [Author Response]

*Essential revisions: 1) The experiments rely on endogenous miRNAs that are thought, but not shown to be present in the S2 cells.*

We did directly show the presence of miRNA 13b and bantam in the cells we used, see Figure 2; perhaps the reviewers overlooked the content of these figures.

*The only data linking particular miRNAs to the effects observed are mutations in the miRNA binding sites. These mutations could alter a variety of 3' UTR mediated regulatory processes. The authors need to demonstrate that miR-2 and Bantam mediate all the decay processes observed by blocking their action with antagomirs.*

We agree that a direct demonstration that the effects we observed were the result of miRNA-mediated action would strengthen the results. Nevertheless, the use of antagomirs to miR-2 family miRNAs was not practical for two reasons. First, the family is comprised of eight members whose only common sequence is the seed itself. Accordingly eight distinct antagomirs would have had to be used simultaneously. Second, inactivation of miR-2 family of miRNAs would lead to derepression of endogenous reaper. Production of reaper proteins would promote apoptosis and thus would complicate any interpretation of results. The same is true for Bantam and hid.

To address the concern we used an alternative strategy first introduced by the Steitz lab. We constructed mimic miRNAs complementary to the nucleotides changed in the mutant reaper and hid MREs. Because of low transfection efficiency, it was not possible to use the stable cell lines. Accordingly we cotransfected mimic miRNAs with the same plasmids used to make the stable cell lines. The presence of mimic miRNAs substantively reduced the half life of reporters containing mutant MREs while a control mimic miRNA did not (Figure 1—figure supplement 2). The determined half lives were not identical to these obtained in the stable cell lines perhaps because of altered kinetics in transiently transfected cells.

*2) The arguments in the paper depend on all the mRNA being on polysomes and thus being the major contributor to the mRNA half-life measured on total RNA. The authors should confirm this by measuring how much mRNA is lost prior to the polysome gradient and how much is pelleted in these gradients. Is there a pool of mRNA sequestered in high molecular weight structures (such as P-bodies) that is not seen on the polysomes but nevertheless contributes to decay rate measurements?*

In response to this concern we did an “accounting experiment” analogous to one we published before. This experiment showed that the vast majority of reporter mRNAs are associated with polysomes and that negligible amounts of these RNAs are present in the pellet of the sucrose gradients (Figure 2—figure supplement 3).

*3) The differential decay of the wild type and mutant RNAs is observed in different stable cell lines for each reporter. This could arise from variation in these cell lines affecting mRNA stability. The authors should assess the stability of reporter and endogenous transcripts in multiple clones of each reporter to control for this. Alternatively, both mutant and wild-type reporters could be measured in the same cell line.*

We appreciate this comment but note that we were not using clonal cell lines. Moreover the experiment with the mimic miRNAs described above rules out the possibility that cell type specific factors could influence mRNA half lives that we determined.

*4) The paper is written from the point of view that there will be one pathway of translational repression and decay induced by miRNAs. The authors should discuss the possibility that different mRNAs could show different pathways, or the different mechanisms could be balanced differently in different systems. Such a discussion would include several references already given, but also papers by Wu* et al. *2010 (PMID:21095586) and Stadler* et al. *2012 (PMID:22855835), showing that in some cases mRNA abundance can be stable while protein expression is reduced by a miRNA. In support of cotranslational deadenylation and decapping, they might also reference Antic* et al. *2015 (PMID:25918245).*

We appreciate this comment and note that in the revised Discussion we cite several examples from the literature that are in apparent conflict with our one pathway model. We also cite Antic et al. whose results are totally compatible with ours. We cannot at present explain the apparent translational inhibition observed by others for some specific miRNAs. Time and future experiments will tell if these outliers are meaningful.